# DARE: Disentanglement-Augmented Rationale Extraction

**Linan Yue**[1,2], **Qi Liu**[1,2]*, **Yichao Du**[1,2], **Yanqing An**[1,2], **Li Wang**[1,2,3], **Enhong Chen**[1,2]
1: Anhui Province Key Laboratory of Big Data Analysis and Application,
University of Science and Technology of China
2: State Key Laboratory of Cognitive Intelligence
3: ByteDance
{lnyue,duyichao,anyq,wl063}@mail.ustc.edu.cn;
{qiliuql,cheneh}@ustc.edu.cn

## Abstract

Rationale extraction can be considered as a straightforward method of improving the model explainability, where rationales are a subsequence of the original inputs, and can be extracted to support the prediction results. Existing methods are mainly cascaded with the *selector* which extracts the rationale tokens, and the *predictor* which makes the prediction based on selected tokens. Since previous works fail to fully exploit the original input, where the information of non-selected tokens is ignored, in this paper, we propose a Disentanglement-Augmented Rationale Extraction (DARE) method, which encapsulates more information from the input to extract rationales. Specifically, it first disentangles the input into the rationale representations and the non-rationale ones, and then learns more comprehensive rationale representations for extracting by minimizing the mutual information (MI) between the two disentangled representations. Besides, to improve the performance of MI minimization, we develop a new MI estimator by exploring existing MI estimation methods. Extensive experimental results on three real-world datasets and simulation studies clearly validate the effectiveness of our proposed method. Code is released at `https://github.com/yuelinan/DARE`.

## 1 Introduction

Although the performance of deep neural networks (DNNs) has significantly improved across a range of natural language understanding tasks [43, 49, 50], the inability of DNNs to provide the explainability for their predictions still remains a serious risk. To this end, several researchers [35, 26, 28] have focused on improving the explainability of DNNs. Among them, the rationale extraction method [24, 4] can provide an intuitive explanation by identifying the important features of inputs. Specifically, it extracts a short and coherent part of original inputs (i.e., the *rationale*) as an explanation to support the prediction results when yielding them.

Traditional rationale extraction approaches [24, 4, 44, 33] cascade the *selector* and the *predictor*. As shown in Figure 1(a), we illustrate this type of approaches with an example of the charge prediction, where charges are automatically predicted based on the case fact. Specifically, the *selector* first selects a subsequence of the fact description (i.e., the bolded tokens in Figure 1(a). Then, the *predictor* yields the charge result based on the selected tokens. And the extracted subsequence is defined as the rationale. However, since the extractive rationales only depend on the comparison between the prediction results and labels, this kind of approach may fail to fully exploit the information of the original input. To solve that, Sha et al. [37] add an external *guider* (i.e., Figure 1(b)) which takes the

---

*Corresponding Author

36th Conference on Neural Information Processing Systems (NeurIPS 2022).

whole text as the input and generate the accurate but uninterpretable representations as most DNNs do to predict the result. Then, the *guider* utilizes the above representations to guide the *predictor* to yield more comprehensive task-related representations with an adversarial-based method. Since this method fails to utilize the information of the original text, where the non-rationale tokens are ignored, we argue that this "*guidance pattern*" can be further explored to improve the rationale extraction.

Along this research line, in this paper, we propose a " *self-guided* " method, **D**isentanglement-**A**ugmented **R**ationale **E**xtraction (DARE), which augments traditional rationale extraction approaches with the disentangled representations learning. Different from the previous model that requires external guidance, DARE aims to guide itself to extract more comprehensive rationales by squeezing more information from the input. Specifically, we first disentangle the original inputs into two parts: the rationale tokens and the non-rationale ones by the *selector*, where the two type tokens should be independent ideally. Then, in the *predictor*, consisting of an encoder and a classifier, these two type tokens are passed into a shared encoder to obtain the corresponding representations. Finally, DARE utilizes rationale representations to yield results. Meanwhile, the *guider* adopts non-rationale representations to guide the rationale ones to be more comprehensive by reducing the dependency between the two representations (i.e., minimizing their Mutual Information (MI)).

Besides, to improve the performance of minimizing MI, based on the state-of-the-art MI minimization method, Contrastive Log-ratio Upper Bound (CLUB) [11, 12], we propose a new implementation of CLUB (i.e., CLUB_NCE) by exploring

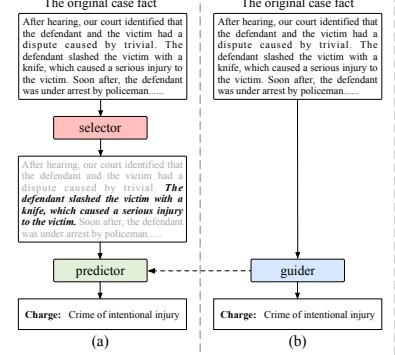

Figure 1: Schematic of existing rationale extraction approaches presented in this paper. (a) illustrates a traditional process of rationale extraction that utilizes the selected rationale (in **bold**) to yield results. (b) shows an external guider to guide the rationale selection based on the whole input.

the relationship between CLUB and InfoNCE (a classical maximizing MI method [32]). Following the experimental setup of [34, 11], we evaluate the effectiveness of CLUB_NCE on two simulation studies by comparing with other MI minimization methods.

To validate the effectiveness of our proposed DARE, we perform extensive experiments on the BeerAdvocate dataset [29] which is utilized to the multi-aspect sentiment analysis, a movie review dataset [48], and a legal judgment prediction dataset (CAIL2018) [41] which is adopted to predict judgment results including charges, law articles and terms of penalty. The experimental results empirically show that DARE has achieved better performance on both extracting rationales and predicting task results compared to other state-of-the-art rationale extraction methods.

## 2 Disentanglement-Augmented Rationale Extraction

### 2.1 Problem Definition

Given a text input $x = \{x_1, x_2, \ldots, x_n\}$ consisting of $n$ tokens and the ground truth $y$, our goal is first to learn a mask variable $m = \{m_1, m_2, \ldots, m_n\}$, and then learn a model which adopts the rationale $z = m \odot x = \{m_1 \cdot x_1, m_2 \cdot x_2, \ldots, m_n \cdot x_n\}$ to yield the prediction results. Take the case in Figure 1 for example, based on the fact description $x$, DARE aims to yield the charge result $y$ while extracting the rationale $z$ to support this result. Next, we will present the details of our proposed " *self-guided* " method, DARE, which consists of the *selector*, *predictor* and *guider*.

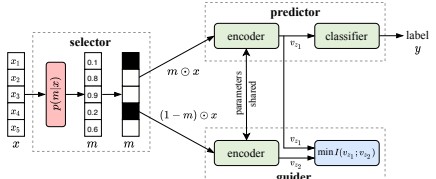

Figure 2: Architecture of DARE consisting of the selector, predictor and guider : the white boxes indicate the rationales related tokens and the black are non-related ones.

### 2.2 Architecture of DARE

To utilize the non-rationale representations to enhance the rationale ones, and further extract more accurate rationales, we propose the DARE method which is shown in Figure 2. In DARE, we first

utilize the *selector* to select the rationale tokens and the non-rationale ones, and adopt the shared encoder to generate the corresponding representations. Then, the classifier in the *predictor* can yield the prediction results based on the rationale representations. Meanwhile, the *guider* considered as a regularizer minimizes the MI between the above two types of representations to ensure they are explicitly disentangled, which can make the rationale representations more comprehensive.

### 2.2.1 Selector

To select the rationale tokens, the *selector* takes $x = \{x_1, x_2, \ldots, x_n\}$ as the input to generate a binary variable $m = \{m_1, m_2, \ldots, m_n\}$, and adopts $m$ to sample the rationale and non-rationale tokens. Specifically, we first utilize the *selector* to generate a probability distribution $p(m|x)$ which represents the probability of sampling each $x_i$ as the part of the rationale. To ensure this operation of sampling is differentiable, we adopt HardKuma reparameterization trick [4] to generate the $p(m|x)$:

$$k_j = F_{Kuma}^{-1}(u_j; a_j, b_j), t_j = l + (r - l) \times k_j, p(m_j|x) = \min\left(1, \max\left(0, t_j\right)\right), \quad (1)$$

where $u_j$ is random sampled from the uniform distribution $U(0, 1)$, $F_{Kuma}^{-1}(u; a, b) = \left(1 - (1 - u)^{1/b}\right)^{1/a}$ is the inverse c.d.f. of the Kumaraswamy distribution [23], $a$ and $b$ are parameterized by DNNs which takes $x$ as the input, and $r$ and $l$ are fixed hyperparameters. Finally, we can derive the rationale tokens as $z_1 = m \odot x$ and the non-rationale ones as $z_2 = (1 - m) \odot x$.

### 2.2.2 Predictor

The *predictor* $q_\psi(y|z_1)$ which consists of an encoder and a classifier outputs the prediction results based on the rationale tokens $z_1$. Specifically, we first re-embed the $z_1$ into continuous hidden states by the encoder to obtain the rationale representations $v_{z_1} \in \mathbb{R}^d$, where $d$ is the hidden size. Then, we pass the $v_{z_1}$ into the classifier to yield the results $y$ which is predicted based exclusively on the rationale tokens extracted by *selector*, and the task loss function can be formulated as :

$$\mathcal{L}_{task} = \mathbb{E}_{z_1 \sim p_\sigma(z_1|x)}\left[-\log q_\psi(y \mid z_1)\right], \quad (2)$$

where $p_\sigma$ is the neural network of *selector*.

### 2.2.3 Guider

While predicting results, to guide the model itself to extract more comprehensive rationales, we augment the disentangled representations learning (DRL) into the *guider*. Specifically, we pass non-rationale tokens $z_2$ to the shared encoder described in section 2.2.2 to obtain the non-rationale representations $v_{z_2}$. Then, we hope $v_{z_1}$ and $v_{z_2}$ should be independent ideally so that the $v_{z_1}$ can contain rich rationale information. To achieve this goal, we propose the mutual information minimization method which reduces the dependency between $v_{z_1}$ and $v_{z_2}$ to disentangle them (i.e., min $I(v_{z_1}; v_{z_2})$). Besides, following the application of MI in DRL [12, 36, 20], we should maximize $I(v_x; v_{z_1}, v_{z_2})$ (a reconstruction loss) to ensure $v_{z_1}$ and $v_{z_2}$ encapsulate information from the input $x$ sufficiently, where $v_x$ is the original input representations. However, since $v_{z_1}$ and $v_{z_2}$ are obtained from the separate input, we argue that $v_{z_1}$ and $v_{z_2}$ have preserved the original input itself, and can remove this regularization. In section 2.3, we will describe the MI minimization method in detail.

## 2.3 Mutual Information Minimization

In this section, we first review some concepts of MI, and then present a new MI minimization method (i.e., CLUB_NCE) by discussing the relationships between InfoNCE and CLUB.

### 2.3.1 Mutual Information

In probability theory and information theory, the mutual information (MI) of two random variables (e.g., $X$ and $Y$) is a measure of the mutual dependence between the two variables $I(X; Y) = \mathbb{E}_{p(x,y)}\left[\log \frac{p(x,y)}{p(x)p(y)}\right]$. However, directly computing MI values is difficult [5], especially when $X$ and $Y$ are continuous and high-dimensional. To approximate MI, for MI maximization tasks, Oord et al. [32] derive a lower-bound MI estimation (InfoNCE):

$$I_{nce} = \frac{1}{N} \sum_{i=1}^{N} \log \frac{e^{f(x_i, y_i)}}{\frac{1}{N} \sum_{j=1}^{N} e^{f(x_i, y_j)}} = \frac{1}{N} \sum_{i=1}^{N} f(x_i, y_i) - \frac{1}{N} \sum_{i=1}^{N} \left[\log \frac{1}{N} \sum_{j=1}^{N} e^{f(x_i, y_j)}\right], \quad (3)$$

where $\{(x_i, y_i)\}_{i=1}^N$ is a batch of sample pairs and $f(x, y)$ is a learnable score function. For MI minimization tasks, Cheng et al. [11, 12] propose a Contrastive Log-ratio Upper Bound (CLUB) method:

$$I_{club} = \frac{1}{N} \sum_{i=1}^N \log p(y_i|x_i) - \frac{1}{N^2} \sum_{i=1}^N \sum_{j=1}^N \log p(y_j|x_i), \quad (4)$$

where $p(y|x)$ is a conditional distribution.

### 2.3.2 CLUB_NCE

Although $p(y|x)$ in CLUB can be any neural network, intuitively, an easy way to parameterize $p(y|x)$ is with an MLP. In practical, $p(y|x)$ is always parameterized by a Gaussian family[2], where one potential reason for not employing MLP is that it is difficult to converge in CLUB. Therefore, it is significantly to design a function that can parameterize $p(y|x)$ with any network and be easy to converge. To this end, we first discuss the relationship between InfoNCE and CLUB. Specifically, by applying the Jensen's inequality in the Eq (3), we can get:

$$I_{nce} \leq \frac{1}{N} \sum_{i=1}^N f(x_i, y_i) - \frac{1}{N^2} \sum_{i=1}^N \sum_{j=1}^N \log \left[ e^{f(x_i, y_j)} \right] = \frac{1}{N} \sum_{i=1}^N f(x_i, y_i) - \frac{1}{N^2} \sum_{i=1}^N \sum_{j=1}^N f(x_i, y_j),$$
$$(5)$$

where we note this inequality is similar to Eq (4) and the main difference is the score function. Interestingly, we find for any critic $f(x, y)$ in InfoNCE, the optimal critic is $f^*(x, y) = \log(p(y \mid x))$ [34] which is equivalent to the critic function in Eq (4).

Inspired by this observation, we combine CLUB with InfoNCE empirically and propose a new implementation of CLUB (i.e., CLUB_NCE) to minimize MI, which first adopts the trained $f(x, y)$ by InfoNCE to replace the $\log(p(y \mid x))$ in CLUB to calculate the value of $I_{club}$, and then minimizes it to minimize MI.

We illustrate the process of MI Minimization with CLUB_NCE in Algorithm 1. At each training iteration, we first sample $\{(x_i, y_i)\}_{i=1}^N$ from the generator $g_\phi(x, y)$. Then, we update the $f_\theta(x, y)$ by maximizing $I_{nce}$. Next, the value of $I_{club}$ can be calculated as described in Eq (5) and $g_\phi(x, y)$ can be updated. The neural network $f_\theta$ and $g_\phi$ are trained alternately. Specifically, in DARE, we set the shared encoder as the $g_\phi$, and the *guider* which is parameterized with a full connection layer as the $f_\theta$.

---

**Algorithm 1** MI Minimization with CLUB_NCE

**for** each training iteration **do**
    Sample $\{(x_i, y_i)\}_{i=1}^N$ from $g_\phi(x, y)$.
    Compute the InfoNCE loss as:
    $I_{nce} = \frac{1}{N} \sum_{i=1}^N \log \frac{\exp(f_\theta(x_i, y_i))}{\frac{1}{N} \sum_{j=1}^N \exp(f_\theta(x_i, y_j))}$.
    Update the $f_\theta(x, y)$ by maximizing $I_{nce}$.
    **for** $i = 1$ ; $i <= N$ ; $i++$ **do**
        $U_i = f_\theta(x_i, y_i) - \frac{1}{N} \sum_{j=1}^N f_\theta(x_i, x_j)$.
    **end for**
    Update the $g_\phi(x, y)$ by minimizing :
    $I_{club} = \frac{1}{N} \sum_{i=1}^N U_i$.
**end for**

---

### 2.4 Training

During training, following the setup of [4], we first penalize $\mathcal{L}_{sh} = \frac{1}{n} \sum_{j=1}^n [1 - p(m_j = 0|x)]$ to ensure the model extracts short rationales, where $p(m_j = 0|x)$ is the probability of the token not being selected. Then, we add a regularizer to encourage coherence of selected tokens[3]:

$$\mathcal{L}_{re} = \frac{1}{n-1} (\sum_{j=1}^{n-1} p(m_j = 0 \mid x)(1 - p(m_{j+1} = 0 \mid x)) + (1 - p(m_j = 0 \mid x)) p(m_{j+1} = 0 \mid x).$$
$$(6)$$

By combining Eq (2), $\mathcal{L}_{sh}$, $\mathcal{L}_{re}$ and the $I_{club}$ calculated by CLUB_NCE between $v_{z_1}$ and $v_{z_2}$, our final objective of DARE is defined as $\mathcal{L} = \lambda_1 \mathcal{L}_{task} + \lambda_2 |\mathcal{L}_{sh} - l_r| + \lambda_3 \mathcal{L}_{re} + \lambda_4 I_{club}$, where $\lambda_1$ and $\lambda_4$ are fixed hyperparameters, $\lambda_2$ and $\lambda_3$ are Lagrangian multipliers which will be updated while training, and $l_r$ represents that how many tokens DARE will select.

---

[2] https://github.com/Linear95/CLUB
[3] For more details of $\mathcal{L}_{sh}$, $\mathcal{L}_{re}$ and HardKuma, refer to [4].

Table 1: Precision, Recall and F1 of selected rationales for three aspects. Among them, "% selected" represents the average proportion of selected tokens in the original text.

| Methods | Appearance | | | | Smell | | | | Palate | | | |
|---|---|---|---|---|---|---|---|---|---|---|---|---|
| | Precision | Recall | F1 | % selected | Precision | Recall | F1 | % selected | Precision | Recall | F1 | % selected |
| Bernoulli | 96.3 | 56.5 | 71.2 | 14 | 95.1 | 38.2 | 54.5 | 7 | 80.2 | 53.6 | 64.3 | 7 |
| HardKuma | **98.1** | 65.1 | 78.3 | 13 | **96.8** | 31.5 | 47.5 | 7 | **89.8** | 48.6 | 63.1 | 7 |
| InfoCal_IB | 97.3 | 67.8 | 79.9 | 13 | 94.3 | 34.5 | 50.5 | 7 | 89.6 | 51.2 | 65.2 | 7 |
| InfoCal(HK) | 97.9 | 71.7 | 82.8 | 13 | 94.8 | 42.3 | 58.5 | 7 | 89.4 | 56.9 | 69.5 | 7 |
| DARE (L1Out) | 91.5 | 26.7 | 41.3 | 13 | 84.0 | 38.0 | 52.3 | 7 | 55.4 | 57.0 | 56.2 | 7 |
| DARE (CLUB) | 93.7 | 73.0 | 82.1 | 13 | 90.9 | 42.9 | 58.3 | 7 | 88.7 | 54.3 | 67.4 | 7 |
| DARE | 95.1 | **73.5** | **82.9** | 13 | 88.6 | **46.8** | 61.2 | 7 | 85.6 | **59.0** | **69.9** | 7 |
| (std) | ±0.2 | ±0.3 | ±0.1 | - | ±0.8 | ±0.6 | ±0.6 | - | ±0.6 | ±0.5 | ±0.2 | - |

# 3 Experiments

In this section, we first compare DARE with some baselines on a beer reviews dataset, a movie reviews dataset, and a legal judgment prediction dataset. Then, to demonstrate the effectiveness of CLUB_NCE, we compare it with some classical MI minimization methods on both simulation studies and real-world datasets.

## 3.1 Beer Reviews Prediction

Beer Reviews Prediction is formulated as a multi-aspect sentiment analysis task, which predicts the ratings (on a scale of 0 to 5 stars) for multiple aspects (e.g., smell, palate) based on the beer reviews. We use the BeerAdvocate [29] dataset containing more than 220,000 beer reviews as our dataset. For the fair comparison, we replicate the pre-process of [24] and [37] where the dataset has been divided into three aspects including appearance, smell, and palate. We normalize the ratings to $[0, 1]$ and adopt them as the ground truth for the regression. For testing, we take 994 reviews for three aspects as our test set, where the aspect-related rationales are annotated by human.

### 3.1.1 Comparison Methods and Experimental Setup

• **Bernoulli** [24] generates rationales by yielding the Bernoulli distribution of each token.

• **HardKuma** [4] proposes a HardKuma distribution for reparameterized gradient estimates while selecting rationale tokens.

• **InfoCal** [37] introduces an adversarial-based method to guide the selector-predictor model to extract rationales. Since InfoCal proposed a new LM regularizer to encouraging the consecutiveness of rationales, for a fair comparison, we adopt one of the versions of InfoCal (i.e., **InfoCal (HK[4])** in [37]), which applies the same regularizers $\mathcal{L}_{sh}$ and $\mathcal{L}_{re}$ to ensure the rationales are short and coherence.

• **InfoCal_IB** [37] adopts an Information Bottleneck regularizer to manage the trade-off between yielding short rationales and accurate results by removing the adversarial module in InfoCal.

We replicate the setup of [37] where the architecture of *selector* and *encoder* are RCNN and the loss function is the mean-squared error (MSE) loss. Besides, we adopt the precision, recall, and F1-score to evaluate the performance of the selected rationales. Among them, the precision is defined as how many the selected tokens are in annotated rationales, and the recall represents how many annotated rationale tokens are selected. For a fair comparison, we set the $l_r$ as $\{0.13, 0.07, 0.07\}$ in three aspect datasets, respectively, and initialize the epoch as 50, which are both consistent with the previous methods [4, 37]. Several results of baselines in Table 1 are directly taken from [37].

### 3.1.2 Experimental Results

To demonstrate the effectiveness of DARE, we compare it with baselines on the beer review prediction task, and experimental results are shown in Table 1. From the results, we can find DARE achieves better performance on the three aspects of BeerAdvocate datasets, especially on recall and F1 scores, which demonstrates that DARE can select more comprehensive rationales. Specifically, compared with the traditional *selector-predictor* models (i.e., Bernoulli, HardKuma and InfoCal_IB), DARE significantly outperforms them, indicating the effectiveness of our method on utilizing the information

---

[4]HK represents adopting the regularizers (i.e., $\mathcal{L}_{sh}$ and $\mathcal{L}_{re}$) in HardKuma.

of non-rationale tokens, where this type tokens are ignored in previous models. As the *selector* and *predictor* in DARE are based on HardKuma, we can consider HardKuma is a variant of DARE, which removes the disentanglement operation. Our model performs better than it, we can conclude that our disentanglement augmented method with MI minimization is effective.

Besides, we analyze the experimental results between DARE and InfoCal(HK) which is also a "guided" model but with an external guider (i.e., Figure 1(b)). DARE does not outperform the InfoCal(HK) on all metrics and some results are close (e.g., F1 in Palate prediction). This is probably because InfoCal(HK) exploits the strength of the external *guider*, a black-box neural network that can yield more accurate but uninterpretable representations. However, as a whole, InfoCal(HK) does not perform as well as our model, which demonstrates the effectiveness of squeezing the non-rationale information from input. We also analyze the effectiveness of CLUB_NCE by comparing other MI minimization methods (e.g., CLUB [11] and L1Out [34]), and the results are shown in section 3.4.4.

In the section 2.2.3, we argue that the separable tokens have preserved the information of the original input and DARE does not need the reconstruction loss. To validate this argument, we implement DARE with the reconstruction loss by maximizing $I(v_x; v_{z_1}, v_{z_2})$. Among them, the token F1 scores are **82.6**, **61.4** and **69.8** for appearance, smell and palate, respectively, and so far the results are similar to DARE. Although DARE with reconstruction loss improves on some metrics over DARE, the improvement is very limited, moreover, DARE with reconstruction loss increases the parameters of the model. Therefore, we consider that the reconstruction loss is not necessary.

Since RCNN is used as the *selector-predictor* architecture, to illustrate DARE is agnostic of the model structure, we implement HardKuma and DARE with pretrained models (i.e., adopting BigBird [47] to replace RCNN) on the appearance aspect. The token F1 of HardKuma is **84.4** and DARE is **87.5**, where DARE still outperforms HardKuma, demonstrating the effectiveness of the DARE structure.

## 3.2 Movie Reviews Prediction

Besides the BeerAdvocate dataset, we also make experiments on a movie review dataset [48] of the ERASER benchmark [15], which also contains token-level human rationale annotations. The movie reviews prediction task is formulated as a binary classification, yielding the sentiment of movie reviews (i.e., positive or negative). Table 2 gives results on this task. From the observation, we find DARE achieves the best performance on the three metrics, which further validates the effectiveness of DARE.

Table 2: Results on movie review.

| Methods | Movie | | |
|---|---|---|---|
| | Precision | Recall | F1 |
| Bernoulli | 35.6 | 26.0 | 28.0 |
| HardKuma | 31.1 | 28.3 | 27.0 |
| InfoCal(HK) | **36.7** | 30.3 | 33.2 |
| DARE | 30.7 | **36.6** | **33.4** |
| (std) | ±3.1 | ±2.1 | ±0.7 |

## 3.3 Legal Judgment Prediction

Legal Judgment Prediction (LJP) can be formulated as a text classification problem, which adopts the case fact to yield the judgment results (i.e., charge, law articles, and terms of penalty). We conduct our experiments on publicly available datasets of the **C**hinese **AI** and **L**aw challenge CAIL2018 [41]. CAIL2018 contains criminal cases which consists of fact description and corresponding charges, law articles, and terms of penalty results. Following the data process of [46], we divide the terms into non-overlapping intervals. Then, the LJP task can be formulated as a multi-class classification problem.

In addition to comparing the methods in the rationale extraction, we also compare with some classical baselines in LJP task including FLA [27], TopJudge [51], LADAN [42] and NeurJudge [46]. The above baselines are trained with a multi-task framework which exploits the dependence among the sub-tasks in LJP. Besides, we conduct experiments on one of versions of CAIL2018 (denoted by CAIL) which contains 134,739 cases [46]. We set the $l_r$ as {0.14, 0.14, 0.14} for training. For evaluating, as the CAIL dataset does not contain annotations of rationales, we first adopt the accuracy (Acc), macro-precision (MP), macro-recall (MR), and macro-F1 (F1) to evaluate the performance of yield judgment results. Then, we provide a qualitative analysis on the rationale extraction. Detailed description of comparison methods and experimental setups can be found in Appendix A.

### 3.3.1 Experimental Results

To evaluate the performance of our model on LJP, we show the experimental results from three aspects. First, comparing with the traditional rationale extraction methods, we can find DARE performs well

Table 3: Judgment prediction results on CAIL. Therein, the underlined scores are the state-of-the-art performances in LJP but lacking explainability, and the results in **bold** performs second only to NeurJudge but with explainability. Results of LJP baselines in Table 3 are quoted from [46].

| Methods | Charges | | | | Law Articles | | | | Terms of Penalty | | | |
|---|---|---|---|---|---|---|---|---|---|---|---|---|
| | Acc | MP | MR | F1 | Acc | MP | MR | F1 | Acc | MP | MR | F1 |
| FLA | 84.72 | 83.71 | 73.75 | 75.04 | 85.63 | 83.46 | 73.83 | 74.92 | 35.04 | 33.91 | 27.14 | 24.79 |
| TopJudge | 86.48 | 84.23 | 78.39 | 80.15 | 87.28 | 85.81 | 76.25 | 78.24 | 38.43 | 35.67 | 32.15 | 31.31 |
| LADAN | 88.28 | 86.36 | 80.54 | 82.11 | 88.78 | 85.15 | 79.45 | 80.97 | 38.13 | 34.04 | 31.22 | 30.20 |
| NeurJudge | 88.89 | 86.96 | 85.42 | 85.73 | 89.71 | 86.68 | 83.92 | 84.97 | 41.03 | 39.52 | 36.82 | 36.35 |
| Bernoulli | 85.76 | 82.71 | 78.04 | 79.38 | 86.45 | 81.94 | 75.92 | 77.35 | 38.19 | 33.05 | 32.10 | 30.32 |
| HardKuma | 86.26 | 86.22 | 77.83 | 80.04 | 85.76 | 80.92 | 76.85 | 77.96 | 35.94 | 35.26 | 27.52 | 26.09 |
| InfoCal_IB | 87.14 | 84.86 | 80.70 | 82.11 | 87.58 | 83.55 | 78.49 | 80.13 | 36.60 | 33.12 | 28.20 | 26.82 |
| InfoCal(HK) | 87.85 | 86.79 | 82.53 | 83.87 | 87.65 | 84.88 | 80.09 | 81.67 | **38.69** | 35.68 | 31.61 | **31.21** |
| DARE (L1Out) | 83.20 | 80.92 | 73.74 | 75.10 | 85.89 | 82.76 | 76.54 | 78.04 | 24.09 | 11.76 | 9.14 | 3.72 |
| DARE (CLUB) | 82.87 | 83.19 | 73.17 | 74.84 | 86.48 | 84.14 | 77.05 | 78.83 | 37.45 | 33.41 | 28.99 | 27.88 |
| DARE | **88.29** | **86.58** | **83.29** | **84.36** | **88.51** | **85.62** | **81.35** | **82.66** | 38.43 | **36.77** | **33.25** | 30.65 |
| (std) | ±0.20 | ±0.37 | ±0.34 | ±0.31 | ±0.28 | ±0.40 | ±0.31 | ±0.38 | ±0.22 | ±0.17 | ±0.41 | ±0.27 |

| | |
|---|---|
| **HardKuma** | The People's Procuratorate alleged that ***the defendant Tommy and Bob forcibly had sexual relations with the victim in the room of hotel***. In this regard, the public prosecution agency cited the following evidence: ***capture history***, ......, and statements and explanations of the defendant. After hearing, our court ***identified that the defendant Tommy used violence and verbal threats with others to forcibly have sexual relations with the victim in the hotel room*** ...... |
| **InfoCal(HK)** | The People's Procuratorate alleged that the defendant ***Tommy and Bob forcibly had sexual relations with the victim*** in the room of hotel. In this regard, the public prosecution agency cited the following evidence: capture history, ......, and statements and explanations of the defendant. After hearing, our court identified that the defendant Tommy ***used violence*** and verbal threats with others to ***forcibly have sexual relations with the victim in the hotel room*** ...... |
| **DARE** | The People's Procuratorate alleged that the defendant Tommy and Bob ***forcibly had sexual relations with the victim in*** the room of hotel. In this regard, the public prosecution agency cited the following evidence: capture history, ......, and statements and explanations of the defendant. After hearing, our court identified that the defendant Tommy ***used violence and verbal threats with others to forcibly have sexual relations with the victim in the hotel room*** ...... |

Figure 3: Visualized selective rationales with different methods. The underlined tokens represent the real rationales which is used to support the practical charge (i.e., rape), and the blue is the predicted rationales. More examples can be found in Appendix B.1.

on LJP tasks, which further validates the effectiveness of our proposed model based on the previous beer reviews prediction experimental results. Meanwhile, the analysis of the comparison between CLUB_NCE and other methods of minimizing MI are shown in section 3.4.4.

Then, by comparing with methods in LJP (e.g., NeurJudge), we observe DARE still achieves good overall performance, second only to NeurJudge (the state-of-the-art model in LJP) in most metrics. It is worth noting that these LJP methods are modeled based on the multi-task learning, where the charge, article and term of penalty prediction tasks are related and can enhance each other while DARE is a single-task model (i.e., we need to train a separate model for each task). It demonstrates that DARE can extract effective rationale tokens for prediction. Besides, although these LJP methods achieve promising performance, the judgment results still remain unreliable and difficult to explain. In contrast, DARE can achieve a trade-off between the accuracy and explainability.

As there exist no annotations of rationales in CAIL dataset, we provide a qualitative analysis on rationales extracted by DARE. First, Figure 3 shows an example of the extracted rationale for the charge prediction with different methods, where the underlined tokens represent the practical rationales annotated by human judge. From the observation, DARE can select key rationales which could be adopted to support the prediction results. Besides, comparing with other extracted performance of baselines, DARE can select more rationale tokens, which indicates the strength of utilizing the non-rationale representations.

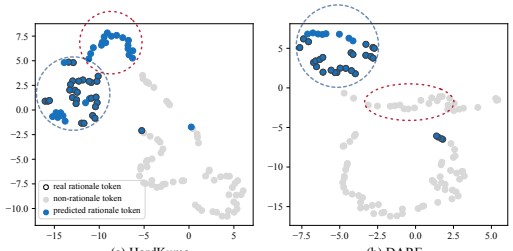

Figure 4: Disentanglement visualizations of rationale representations.

Then, to verify that DARE has effectively disentangled the textual input into the rationale representations and the non-rationale ones, we visualize these representations spaces using t-SNE [40] in Figure 4. Specifically, we visualize the same case (in Figure 3) predicted with HardKuma and DARE. Figure 4(a) shows the visualization of HardKuma. Among them, the blue circle selected represents the tokens that are predicted correctly by HardKuma; and the red circle selected denotes the tokens that are predicted incorrectly, and the position of the red circle is very close to the blue circle. For Figure 4(b), the blue circle selected represents the tokens that are predicted correctly by DARE; and

the red circle selected denotes the tokens that are predicted incorrectly by HardKuma, where our DARE identify that these tokens are non-rationale tokens. From Figure 4(b), we can find the tokens predicted incorrectly by HardKuma are accurately predicted to be non-rationale tokens. It can be seen that after the disentanglement operation, these tokens can be obviously separated and be farther away from the tokens circled in blue. Besides, we calculate the Euclidean(EU) distance between the blue circle and the red circle in Figure 4(a) and 4(b) for measuring their distance. Among them, the EU distance in Figure 4(a) is **7.62**, and in Figure 4(b) is **12.18** which is higher than it in Figure 4(a). The above visualizations demonstrate that DARE can well disentangle rationale representations.

Furthermore, we add a human evaluation to further evaluate extracted rationales in LJP. We sampled 100 examples for the charge prediction task by comparing DARE with HardKuma and InfoCal(HK). Specifically, following [37], we evaluate the rationales with three metrics: usefulness (U), completeness (C), and fluency (F). Among them, each scored from 1 (lowest) to 5 (e.g. 2.0 and 4.9). The scoring standard for human annotators can be found in Appendix A.3. Besides, we ask people to evaluate the extracted rationales and the results are report

Table 4: Human evaluation on charge prediction.

| Methods | Charge | | |
|---|---|---|---|
| | U | C | F |
| HardKuma | 3.95 | 3.30 | 3.45 |
| InfoCal(HK) | 4.45 | 3.78 | 4.25 |
| DARE | **4.58** | **3.92** | **4.32** |

in the table. From the observation on Table 4, we can find DARE outperforms HardKuma and InfoCal(HK) in all metrics, illustrating the effectiveness of DARE.

## 3.4 MI Estimation Quality

In this section, we first evaluate CLUB_NCE on simulation studies with two toy tasks. Then, we show the performance of CLUB_NCE and other MI minimization methods on two real-world applications which described previously.

### 3.4.1 Simulated Studies

Here, we introduce the experimental simulated datasets and comparison methods. First, following the setup from [34, 11], we experiment a dataset where $x$ and $y$ are sampled from a correlated Gaussian distribution: $x \sim \mathcal{N}(0, I_d)$, $y \sim \mathcal{N}\left(\rho x, \left(1 - \rho^2\right) I_d\right)$, where $\rho$ is the correlation and $d$ is the dimensionality. In the first toy task (denoted by **Gaussian**), we sample each dimension of $(x; y)$ from this simulated dataset, and the second task (**Cubic**) is the same as the first while applying the transformation $y \rightarrow y^3$ (i.e., sample $(x; y^3)$ from the same Gaussian distribution). Meanwhile, the value of this mutual information is invariant (i.e., $I(x, y) = I(x, y^3)$) [22]. Under Gaussian distributions, the ground truth of MI is tractable and can be calculated as $I(x, y) = -\frac{d}{2} \log\left(1 - \rho^2\right)$ [34]. For both of tasks, we set the dimension to $d = 20$, hidden size to 128 and batch size to 64. Besides, we set the initial MI value as 2 and increase it by 2 every 4k iterations, for a total of 20k iterations.

Then, we present the comparison methods including InfoNCE [32], CLUB [11], VarUB [1], and L1Out [34]. Among them, CLUB and InfoNCE have been described in section 2.3.1, and VarUB ($I_{varub} = \mathbb{E}_{p(x,y)}\left[\log \frac{p(y|x)}{r(y)}\right]$) and L1Out ($I_{l1out} = \frac{1}{N} \sum_{i=1}^{N}\left[\log \frac{p(y_i|x_i)}{\frac{1}{N-1}\sum_{j \neq i} p(y_i|x_j)}\right]$) are both MI minimization methods which are adopted to estimate the upper bound of MI, where $p(y|x)$ is commonly parameterized by a Gaussian family [11] and $r(y)$ is fixed as a standard normal distribution.

### 3.4.2 Simulated Studies

Here, we introduce the experimental simulated datasets and comparison methods. First, following the setup from Poole et al. [34] and Cheng et al. [11], we experiment a dataset where $x$ and $y$ are sampled from a correlated Gaussian distribution: $x \sim \mathcal{N}(0, I_d)$, $y \sim \mathcal{N}\left(\rho x, \left(1 - \rho^2\right) I_d\right)$, where $\rho$ is the correlation and $d$ is the dimensionality. In the first toy task (denoted by **Gaussian**), we sample each dimension of $(x; y)$ from this simulated dataset, and the second task (**Cubic**) is the same as the first while applying the transformation $y \rightarrow y^3$ (i.e., sample $(x; y^3)$ from the same Gaussian distribution). Meanwhile, the value of this mutual information is invariant (i.e., $I(x, y) = I(x, y^3)$) [22]. Under Gaussian distributions, the ground truth of MI is tractable and can be calculated as $I(x, y) = -\frac{d}{2} \log\left(1 - \rho^2\right)$ [34]. For both of the tasks, we set the dimension to $d = 20$,

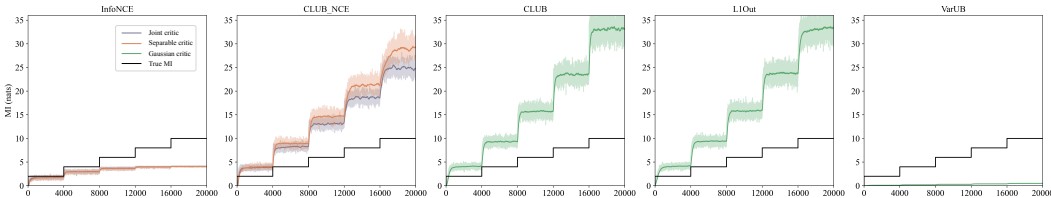

Figure 5: Performance of MI estimation approaches on the **Gaussian** task. Among them, the true MI values are the black line. The estimated values are drawn as light shadow curves while the dark shadow ones represent the local averages of estimated values, with a bandwidth 200.

hidden size to 128 and batch size to 64. Besides, we set the initial MI value as 2 and increase it by 2 every 4k iterations, for a total of 20k iterations.

Then, we present the comparison methods including InfoNCE [32], CLUB [11], VarUB [1], and L1Out [34]. Among them, CLUB and InfoNCE have been described in section 2.3.1, and VarUB ($I_{varub} = \mathbb{E}_{p(x,y)}\left[\log \frac{p(y|x)}{r(y)}\right]$) and L1Out ($I_{l1out} = \frac{1}{N}\sum_{i=1}^{N}\left[\log \frac{p(y_i|x_i)}{\frac{1}{N-1}\sum_{j\neq i}p(y_i|x_j)}\right]$) are both MI minimization methods which are adopted to estimate the upper bound of MI, where $p(y|x)$ is commonly parameterized by a Gaussian family [11] and $r(y)$ is fixed as a standard normal distribution.

### 3.4.3 Experimental Results on Simulated Studies

In this section, we mainly report the experimental analysis on the **Gaussian** task, the analysis on **Cubic** is similar and can be found in Appendix B.2. Figure 5 shows the performance of our CLUB_NCE and other MI estimation approaches. Among them, the joint critic is a single Multilayer Perceptron (MLP) and takes $[x; y]$ as the input, where " ; " represents the concatenate operation, and the separable critic is $f(x, y) = g_1(x)^T g_2(y)$, where $g_1$ and $g_1$ are two different MLPs. The gaussian critic is parameterized by a Gaussian family (i.e.,

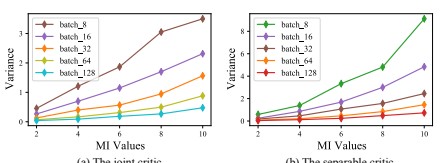

(a) The joint critic    (b) The separable critic

Figure 6: Variance of the CLUB_NCE on the **Gaussian** with different batch sizes in the range {8, 16, 32, 64, 128}.

$\mathcal{N}\left(y \mid \mu(x), \sigma^2(x) \cdot I\right)$), where the mean and variance are calculated by neural networks. We can observe that the InfoNCE values are under the true MI while CLUB_NCE are above it, which illustrates CLUB_NCE is an upper bound of MI. The estimation of VarUB has low value, indicating this method could not remain an upper bound in some cases. The CLUB and L1Out has high bias and variance when the ground truth MI is high, whereas CLUB_NCE has the lower ones, especially the variance. It might have benefited from the strength of InfoNCE which has a low variance.

To further validate the influence of InfoNCE on CLUB_NCE, we first analyze the relationship between them theoretically, and then conduct a comparison among the CLUB_NCE with different batch sizes. Specifically, we review the InfoNCE shown in Eq (3):

$$I_{nce} = \frac{1}{N}\sum_{i=1}^{N}\log \frac{e^{f(x_i,y_i)}}{\frac{1}{N}\sum_{j=1}^{N}e^{f(x_i,y_j)}} = \frac{1}{N}\sum_{i=1}^{N}\log \frac{e^{f(x_i,y_i)}}{\sum_{j=1}^{N}e^{f(x_i,y_j)}} + \log N, \tag{7}$$

where the first term is less than 0 and $I_{nce}$ is upper bounded by $\log N$ [32]. Therefore, the performance of InfoNCE is influenced by batch sizes, where the larger the batch size $N$, the lower variance between the estimated MI and the true one [34]. In other words, since CLUB_NCE adopts the trained critic by InfoNCE (Algorithm 1), the value of $I_{club}$ should have lower variance as the batch size increases. Therefore, we conduct an experiment to estimate CLUB_NCE by adopting various batch sizes. As shown in Figure 6, we calculate the variance between the estimated MI values $I_{club}$ and the theoretical MI ones on the **Gaussian** task with different batch sizes. From the observation, we can conclude that CLUB_NCE with larger batch size achieves the lower variance, which supports the previous theoretical analysis.

### 3.4.4 Real-world Applications

We compare CLUB_NCE with CLUB, L1Out and VarUB on beer reviews prediction and legal judgment prediction. Specifically, we replace CLUB_NCE in DARE with other MI minimization

methods, and denote them as DARE (CLUB), DARE (L1Out) and DARE (VarUB). The experimental results are reported in Table 1 and Table 3. Among them, as DARE (VarUB) fails to converge within epochs in the experimental setup, we do not present its result in the table. From the observation, DARE performs the best which indicates our CLUB_NCE is more effective than others methods at minimizing MI on real-world applications. Besides, these variants of DARE fail to improve on some tasks (e.g., DARE (L1Out) in Palate prediction and Terms of Penalty prediction), and are even less effective than other *selector-predictor* models, illustrating it is crucial for DARE to choose a superior MI estimator to effectively minimize the MI.

## 4  Related Work

**Rationale Extraction.** Rationale extraction methods can provide the model explainability by extracting important features of inputs and many researchers [24, 4, 44, 8, 9, 39, 45] have been attracted. Among them, Lei et al. [24] proposed a classical framework for rationales extraction with a selector and predictor. Following this framework, Bastings et al. [4] introduced a HardKuma distribution for reparameterized gradient estimates and facilitated end-to-end differentiability of this framework. Paranjape et al. [33] and Chen et al. [10] managed trade-off between extracting sparse rationales and generating accurate results by the information bottleneck theory. Since this type of framework failed to exploit the information of the original text, several guider methods [37, 7, 19] have been proposed, which adopted the whole text to guide the rationale extraction. Therein, Sha et al. [37] introduced an adversarial-based technique to make the selector-predictor model learn from the guider model. Cao et al. [7] minimized the KL divergence between the two types models to reduce their difference. Although extensive research has been carried out on the rationale extraction, few considered the information of non-rationale tokens containing rich information for extracting rationales.

**Disentangled Representation Learning.** Disentangled representation learning (DRL) [6, 18, 16, 3] maps different aspects of data into independent latent vectors by adding constraints on the embedding space and has attracted much attention. Recently, information theory has been widely adopted in the applications of DRL. Among them, Sanchez et al. [36], Cheng et al. [12] and Huang et al. [20] maximized MI between the original input and its different aspects to retain as much as possible information of the input. And Cheng et al.[11] and Colombo et al. [14] reduced the dependency among the different aspects of the input to make them independent by minimizing the MI.

**Mutual Information.** Mutual information (MI) is adopted to measure the dependency between random variables. Since computing the values of MI is challenging, and many researches utilized the MI as a regularizer in loss functions, and not concerned with its precise value, recent works focused on MI maximization and minimization. Nguyen et al. [31] studied a lower bound of MI based on the MI f-divergence representation for maximizing MI. Belghazi et al. [5] proposed a mutual information estimator (MINE) which estimated the lower bound of MI. Similarly, InfoNCE was introduced by Oord et al. [32], which adopted the negative sampling estimation to maximize MI. For MI minimization, Poole et al. [34] studied an upper bound of MI estimator L1Out based on reviewing the existing variational bounds on MI. And a contrastive log-ratio upper bound (CLUB) was proposed by [12, 11] to minimize MI. Besides, based on the KL divergence and Renyi divergence, Colombo et al. [14] derived a MI estimator on the basis of a variational upper bound.

## 5  Conclusion

In this paper, we proposed a disentanglement-augmented rationale extraction method (DARE) which squeezed more information from the original input by exploiting the non-rationale tokens. To be specific, DARE disentangled the textual input into two parts (i.e., the rationale representations and the non-rationale ones), and minimized the mutual information (MI) between the two disentanglements for extracting more comprehensive rationales. Besides, to improve the performance of minimizing MI, we introduced a new MI minimization method CLUB_NCE based on exploring the relationship between InfoNCE and CLUB. Experimental results on three real-world datasets and a simulated dataset demonstrated the effectiveness of our proposed method.

**Acknowledgements.** This research was partially supported by grants from the National Key Research and Development Program of China (No.2021YFF0901003), the National Natural Science Foundation of China (Grants No.61922073 and U20A20229).

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
