# A  Comparison Methods and Experimental Setup for LJP

## A.1  Comparison Methods

Besides the common methods in rationale extraction including Bernoulli [24], HardKuma [4], InfoCal_IB and InfoCal [37], we also compare with some classical baselines in LJP task:

- **FLA** [27] is an attention-based neural network which interacts the fact with applicable laws.
- **TopJudge** [51] is a topological multi-task learning model which captures the dependencies among charge, law article and term of penalty predictions.
- **LADAN** [42] is an attention-based model which learns the distinguished law articles representations for LJP.
- **NeurJudge** [46] is a circumstance aware method which explores the practical relationship among subtasks in LJP.

The above baselines are all trained with a multi-task framework, which yields the charge, law article and term of penalty cooperatively. Among them, there exist two versions of **NeurJudge** (i.e., NeurJudge and NeurJudge+). Since NeurJudge+ incorporates the external legal knowledge in the LJP task by adopting the GLE module, while DARE yields the results based only on the case fact, we report the results of NeurJudge (denoted by NeurJudge for simplification) for a fair comparison.

## A.2  Experimental Setup

In this section, we present the detailed experimental setup of DARE in LJP. Specifically, following the setup of Yue et al. [46], we employ THULAC [38] for word segmentation, adopt the word2vec [30] for word embedding pre-training with embedding size 200, and implement the *selector* and *encoder* as Bi-GRU [13]. For training, we initialize the learning rate of the Adam optimizer [21] as $10^{-4}$, set the loss function as cross-entropy and the epoch as 16. For evaluating, we employ the accuracy (Acc), macro-precision (MP), macro-recall (MR), and macro-F1 (F1) to evaluate the performance of yielding judgment results including charge, law article and term of penalty.

## A.3  Scoring standards for human evaluation

Following [37], we evaluate the rationales with three metrics: usefulness (U), completeness (C), and fluency (F) in the charge prediction. Among them, each scored from 1 (lowest) to 5. Below, we introduce scoring standards for the above metrics in brief. Detailed standards for human annotators can be found in [37].

### A.3.1  Usefulness

Q: Do you think the selected rationales can be useful for explaining the predicted labels?

- 5: Exactly. Selected rationales are useful for me to get the correct label.
- 4: Highly useful. Although several tokens have no relevance to correct label, most selected tokens are useful to explain the labels.
- 3: Half of them are useful. About half of the tokens are useful for getting labels.
- 2: Almost useless. Almost all of the tokens are useless.
- 1: No Use. The selected rationales are useless for identifying labels.

### A.3.2  completeness

Q: Do you think the selected rationales are enough for explaining the predicted labels?

- 5: Exactly. Selected rationales are enough for me to get the correct label.
- 4: Highly complete. Several tokens related to the label are missing.
- 3: Half complete. There are still some important tokens that have not been selected, and they are in nearly the same number as the selected tokens.
- 2: Somewhat complete. The selected tokens are not enough.
- 1: Nonsense. None of the important tokens is selected.

### A.3.3 fluency

Q: Do you think the selected rationales are fluent?

- 5: Very fluent.
- 4: Highly fluent.
- 3: Partial fluent.
- 2: Very unfluent.
- 1: Nonsense.

# B  More Experimental Results

## B.1  Visualized Selective Rationales

In this section, as there exist no human annotations of rationales in CAIL dataset, we provide more visualization cases in Figure 7 to show the performance of extracting. From the observation, we can find that DARE can select plausible rationales.

## B.2  Experimental Results of Mutual Information

In this section, we show the experimental results on both **Gaussian** and **Cubic** dataset in Figure 8 and Figure 9.

# C  Broader Impact and Limitations.

In this section, we discuss the broader impact and limitations of DARE. Although we construct experiments on text regression and text classification tasks, we briefly argue that our DARE can be potentially extensible to other natural language understanding tasks, such as abstractive Text Generation (TG), extractive Reading Comprehension (RC), Named Entity Recognition (NER). Specifically, the goal of the abstractive text generation task is to select key sentences from the original text. We can apply our DARE to this task by decomposing the input into a sequence of sentences rather than tokens as described in this paper. The extractive reading comprehension aims to extract a continuous segment from the context to answer the question. DARE could utilize the non-answer tokens to help predict the answer. In NER, we can first formulate it as an extractive RC task following Li et al. [25], and then apply DARE used in RC to solve this task.

Besides, we further discuss the possibilities of extending our DARE to other decision-making systems such as the justice system to improve their explainability. In the justice domain, we have applied DARE in the legal judgment prediction (described in section 3.3). Additionally, we can implement DARE in the lawyer proficiency assessment [2] which is assessed based on litigation records. Specifically, we can adopt DARE to the analysis of litigation texts, and extract the rationales which support the prediction of lawyers proficiency, explicitly improving the explainability of the prediction results. So far, no negative impact has been observed.

However, our DARE still suffers from depending on the predefined sparsity $l_r$ in section 2.4 to select tokens. For example, when we set $l_r = 0.13$ in Appearance prediction, it means that DARE is encouraged to select 13% tokens of the original input. Unfortunately, in practice, since the percentage of rationale tokens may be less than 13%, serval non-rationaled tokens will still be selected. To address this problem, we argue that a promising approach is to design an adaptive network [17] to predict the sparsity $l_r$ for each training sample. We will leave it for future work.

| Bernoulli | The People's Procuratorate alleged that the defendant Tommy **and** Bob ***forcibly*** ***had*** ***sexual*** ***relations*** ***with*** ***the*** ***victim*** ***in the room of hotel***. In this regard, the public prosecution agency cited the following evidence: capture history, ......, and statements and explanations of the defendant. After hearing, our court identified that the defendant Tommy used violence and verbal threats with ***others*** ***to forcibly*** ***have*** ***sexual*** ***relations*** ***with*** ***the*** ***victim*** ***in the hotel room*** ...... |
| HardKuma | The People's Procuratorate alleged that ***the defendant Tommy and Bob forcibly*** ***had*** ***sexual*** ***relations*** ***with*** ***the*** ***victim*** ***in the room of hotel***. In this regard, the public prosecution agency cited the following evidence: ***capture history***, ......, and statements and explanations of the defendant. After hearing, our court ***identified that the defendant Tommy used violence and verbal threats with others to forcibly have sexual relations with the victim in the hotel room*** ...... |
| InfoCal_IB | The People's Procuratorate alleged that the defendant ***Tommy and Bob forcibly*** ***had*** ***sexual*** ***relations*** with the victim ***in the room of hotel***. In this regard, the public prosecution agency cited the following evidence: capture history, ......, and statements and explanations of the defendant. After hearing, our court identified that the defendant Tommy used ***violence*** ***and*** ***verbal*** ***threats*** with others ***to*** ***forcibly*** ***have*** ***sexual*** ***relations*** with the victim ***in the hotel room*** ...... |
| InfoCal(HK) | The People's Procuratorate alleged that the defendant ***Tommy and Bob forcibly*** ***had*** ***sexual*** ***relations*** ***with*** ***the*** ***victim*** in the room of hotel. In this regard, the public prosecution agency cited the following evidence: capture history, ......, and statements and explanations of the defendant. After hearing, our court identified that the defendant Tommy ***used*** ***violence*** and verbal threats with others to ***forcibly*** ***have*** ***sexual*** ***relations*** ***with*** ***the*** ***victim in the hotel room*** ...... |
| DARE | The People's Procuratorate alleged that the defendant Tommy and Bob ***forcibly*** ***had*** ***sexual*** ***relations*** ***with*** ***the*** ***victim*** ***in*** the room of hotel. In this regard, the public prosecution agency cited the following evidence: capture history, ......, and statements and explanations of the defendant. After hearing, our court identified that the defendant Tommy ***used*** ***violence*** ***and*** ***verbal*** ***threats*** ***with*** ***others*** ***to*** ***forcibly*** ***have*** ***sexual*** ***relations*** ***with*** ***the*** ***victim*** ***in the hotel room*** ...... |

(a) Visualized selective rationales with different methods on the charge prediction. The real charge in this case is *Rape.*

| Bernoulli | After hearing, our court identified that the defendant was invited by a friend to have dinner with the victim. During this time, the defendant and the victim argued over a toast, and the victim was dissatisfied with the defendant's attitude towards drinking, so ***he verbally abused the defendant***. The defendant then took a folding knife from his battery car at the entrance of the restaurant and ***stabbed*** ***the*** ***victim in*** ***the*** ***chest*** ***and*** ***abdomen*** in the course of the fight. The victim was then taken to hospital for treatment and his injuries were assessed to constitute serious injuries ...... |
| HardKuma | After hearing, our court identified that the defendant was invited by a friend to have dinner with the victim. During this time, the defendant and the victim argued over a toast, and the victim was dissatisfied with the defendant's attitude towards drinking, so ***he verbally abused the defendant***. ***The defendant then*** ***took a folding knife from his battery car*** at the entrance of the restaurant and ***stabbed*** ***the*** ***victim in the*** ***chest*** ***and abdomen*** ***in*** ***the*** ***course*** ***of*** ***the*** ***fight***. The victim was then taken to hospital for treatment and his injuries were assessed to constitute serious injuries ...... |
| InfoCal_IB | After hearing, our court identified that the defendant was invited by a friend to have dinner with the victim. During this time, the defendant and the victim argued over a toast, and the victim was dissatisfied with the defendant's attitude towards drinking, so he verbally ***abused the defendant***. The defendant then ***took*** ***a*** ***folding*** ***knife*** from his battery car at the entrance of the restaurant and ***stabbed*** ***the*** ***victim*** ***in*** ***the*** ***chest*** and abdomen in the course of the fight. The victim was then taken to hospital for treatment and his injuries were assessed to constitute ***serious*** ***injuries*** ...... |
| InfoCal(HK) | After hearing, our court identified that the defendant was invited by a friend to have dinner with the victim. During this time, the defendant and the victim argued over a toast, and the victim was dissatisfied with the defendant's attitude towards drinking, so ***he verbally abused the defendant***. ***The defendant*** then ***took*** ***a folding*** ***knife*** from his battery car at the entrance of the restaurant and ***stabbed*** ***the*** ***victim*** ***in*** ***the*** ***chest*** ***and*** ***abdomen*** ***in*** ***the*** ***course*** ***of*** ***the*** ***fight***. The victim was then taken to hospital for treatment and his injuries were assessed to constitute ***serious*** ***injuries*** ...... |
| DARE | After hearing, our court identified that the defendant was invited by a friend to have dinner with the victim. During this time, the defendant and the victim argued over a toast, and the victim was dissatisfied with the defendant's attitude towards drinking, so he verbally abused the defendant. ***The defendant*** ***then*** ***took*** ***a folding*** ***knife from his battery car*** at the entrance of the restaurant ***and*** ***stabbed*** ***the*** ***victim*** ***in*** ***the*** ***chest*** ***and*** ***abdomen*** ***in*** ***the*** ***course*** ***of*** ***the*** ***fight***. The victim was then taken to hospital for treatment and his injuries ***were*** ***assessed*** ***to*** ***constitute*** ***serious*** ***injuries*** ...... |

(b) Visualized selective rationales with different methods on the law article prediction. The real article in this case is *Law article 234.*

| Bernoulli | After hearing, our court identified that at 15:10 on March 2, 2017, the defendant ***seized*** ***the*** ***mobile*** ***phones*** ***away*** when the employee was not prepared in the name of buying a mobile phone at the mobile phones store. Soon after, the defendant was under arrest by policemen. After identification, ***the*** ***value*** ***of*** ***phones*** ***involved*** ***was*** ***1,200*** ***dollars***. Besides, the defendant committed the crime because he was broke and unable to afford the treatment for his mother. |
| HardKuma | After hearing, our court identified that at 15:10 on March 2, 2017, the defendant ***seized*** ***the*** ***mobile*** ***phones*** ***away*** ***when*** ***the*** ***employee*** ***was*** ***not*** ***prepared*** ***in the*** ***name of buying a mobile phone at the mobile phones store***. Soon after, the defendant was under arrest by policemen. After identification, ***the*** ***value*** ***of*** ***phones*** ***involved*** ***was*** ***1,200*** ***dollars***. Besides, the defendant committed the crime because he was broke and unable to afford the treatment for his mother. |
| InfoCal_IB | After hearing, our court identified that at 15:10 on March 2, 2017, the defendant ***seized*** ***the*** ***mobile*** ***phones*** ***away*** ***when*** ***the*** ***employee*** ***was*** ***not*** ***prepared*** ***in the*** ***name of buying a mobile phone*** at the mobile phones store. Soon after, the defendant was under arrest by policemen. After identification, ***the*** ***value*** ***of*** ***phones*** ***involved*** ***was*** 1,200 ***dollars***. Besides, the defendant committed the crime because he was broke and unable to afford the treatment for his mother. |
| InfoCal(HK) | After hearing, our court identified that at 15:10 on March 2, 2017, the defendant ***seized*** ***the*** ***mobile*** ***phones*** ***away*** ***when*** ***the*** ***employee*** ***was*** ***not*** ***prepared*** ***in the*** ***name of buying a mobile phone*** at the mobile phones store. Soon after, the defendant was under arrest by policemen. After identification, ***the*** ***value*** ***of*** ***phones*** ***involved*** ***was*** ***1,200*** ***dollars***. Besides, the defendant committed the crime because ***he*** ***was*** ***broke*** ***and*** unable to afford the treatment for his mother. |
| DARE | After hearing, our court identified that at 15:10 on March 2, 2017, the defendant ***seized*** ***the*** ***mobile*** ***phones*** ***away*** ***when*** ***the*** ***employee*** ***was*** ***not*** ***prepared*** ***in the*** ***name of buying a mobile phone*** at the mobile phones store. Soon after, the defendant was under arrest by policemen. After identification, ***the*** ***value*** ***of*** ***phones*** ***involved*** ***was*** ***1,200*** ***dollars***. Besides, the defendant committed the crime because ***he*** ***was*** ***broke*** ***and*** ***unable*** ***to*** ***afford*** ***the*** ***treatment*** for his mother. |

(c) Visualized selective rationales with different methods on the term of penalty prediction. The real term of penalty in this case is *A fixed-term imprisonment of two years.*

Figure 7: A visualized performance of extracted rationales with different methods. Among them, the underlined tokens represents the real rationales, which is used to support the practical judgment results, and the blue is the predicted rationales.

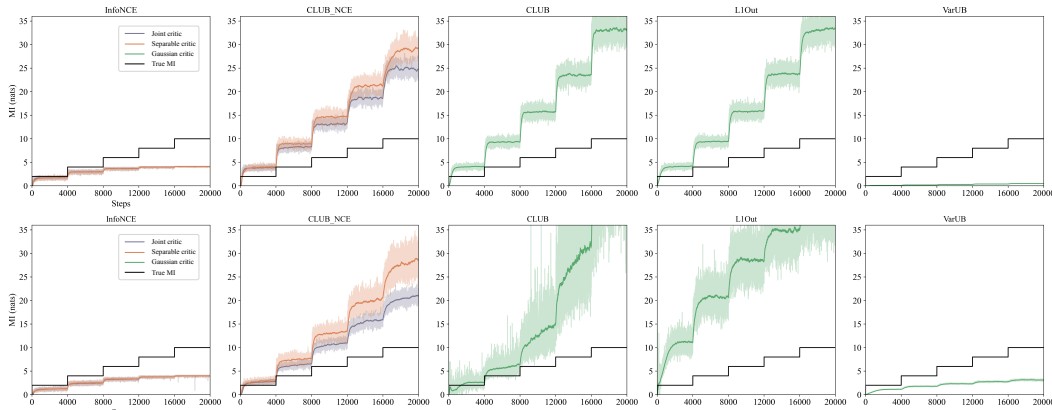

Figure 8: Performance of MI estimation approaches on the **Gaussian** (the top) and **Cubic** (the bottom) tasks. Among them, the true MI values are the black line. The estimated values are drawn as light shadow curves while the dark shadow ones represent the local averages of estimated values, with a bandwidth 200.

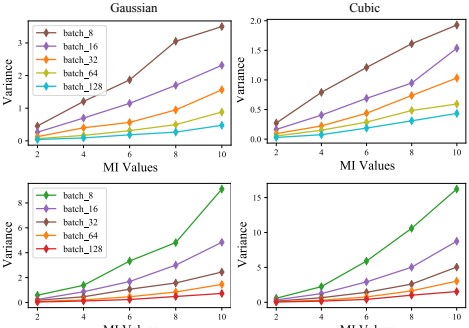

Figure 9: Variance of the CLUB_NCE on the **Gaussian** and **Cubic** task with different batch sizes in the range {8, 16, 32, 64, 128}. The top shows the performance of the joint critic on the these two datasets, and the bottom represents the results of the separable critic.