# OpenReview forum: "DARE: Disentanglement-Augmented Rationale Extraction"
_NeurIPS.cc/2022/Conference — NeurIPS 2022 Accept_

### Official Review · Reviewer_GAz6 · 2022-07-09

**Rating:** 5
**Confidence:** 4
**Soundness:** 2 fair
**Presentation:** 3 good
**Contribution:** 2 fair

**Summary:**

This paper studies a method for rationale extraction for natural language processing (NLP), focusing on text classification tasks. The overall architecture involves a selector to select rationale tokens from an input text, a predictor to consume the selected rationale tokens to obtain a representation for predictions, and a guider to obtain a representation for the non-rationale tokens that will be regularized to minimize the mutual information (MI) with the rationale token-based representation. Compared to previous work, this paper argues that it is helpful to use information for non-rationale tokens to improve the selection of rationale tokens. In addition, to minimize mutual information between two representations, the paper presents a new approach (CLUB_NCE) that combines two existing methods of InfoNCE and CLUB. Experiments are conducted over two text classification datasets to demonstrate better prediction performance and more reasonable rationale for the method. The paper also provides a simulated experiments to compare the proposed MI minimization method with previous baselines and a visualization to demonstrate the disentangled representations for rationale and non-rationale tokens.

**Questions:**

Please consider the weakness presented above.

**Limitations:**

The paper includes a discussion on broader impact and limitations in the appendix. However, the limitation discussion is limited to only selection of the l_r parameter. It might be helpful to further consider the limitations as described in the weakness section.

**Strengths And Weaknesses:**

**Strength

+The paper considers an important problem of rationale extraction to improve interpretability for NLP models.

+The idea of using non-rationale information to improve rationale extraction is interesting and the introduced MI minimization method demonstrates good empirical results.

+The proposed method achieves competitive performance on two datasets that can provide useful baselines for future research.

**Weakness

-While interesting, the guider component (with non-rationale tokens) and the CLUB_NCE for MI minimization seem to be incremental extension of existing methods for rationale extraction and MI estimation. The method mostly involves modification and combination of closely related techniques (e.g., InfoNCE and CLUB).

-As indicated in the Appendix, the model only uses word2vec embeddings to encode texts and does not consider current pre-trained language models (e.g., BERT) that has significantly advanced different NLP tasks. The key question is whether the proposed method can still maintain its claimed benefits when applied with pre-trained language models. In addition, this important implementation information with word2vec and Bi-GRU should be presented in the main paper to make it easier to understand the methods.

-Although Section 3.3 provides empirical experiments to compare different MI estimation methods, it is helpful to better motivate for the designed MI method CLUB_NCE. In particular, Section 2.3.2 mainly describes the implementation of the method. Without a discussion on limitations of previous methods and motivation to address the limitations, it is difficult to understand the benefits of the designed method.

-The performance of the proposed method is very close and sometime worse than the closely related baseline InfoCal(HK), making the effectiveness of the method less convincing. It might be helpful to provide significant scores to strengthen the results.

-Due to the coherence regularization, the selected rationale tokens tend to form consecutive sequences, causing remaining non-selected tokens to involve nonconsecutive words that might lead to unnatural texts. As such, the learned representations for rationale and non-rationale tokens might be quite different, making the MI minimization less reasonable. As such, is it better to obtain non-rationale representation by subtracting rationale representation from the representation for the entire input text to avoid unnatural inputs? It might be helpful to explore this approach to better justify the design choice of the model in this work.

---

> ### Author Response · Authors · 2022-07-31
> **Response to reviewer GAz6**
>
> We appreciate your comments! To address your concerns,  we provide the responses and conduct additional experiments :
>
> **1.Motivation of combining InfoNCE and CLUB.**
>
> $p(y|x)$  in CLUB can be any neural network. Intuitively, an easy way to parameterize $p(y|x)$ is with an MLP. However, in practice,  $p(y|x)$ is always parameterized by a Gaussian family [1,2,3], where one potential reason for not employing MLP is that MLP is difficult to converge in CLUB. Therefore, it is significant to design a function that can parameterize $p(y|x)$  with any network and be easy to converge.  As the critic function in InfoNCE can be any network, and the optimal critic $\log ({p(y \mid x)})$ is equivalent to the critic function in CLUB. Empirically, we combine CLUB with InfoNCE.
>
> **2.Discussion on pre-trained language models.**
>
> We argue that DARE can maintain benefits when applied with pre-trained language models. We report the results in Appearance with HardKuma and DARE, which both implement BigBird [4] to replace RCNN.
>
> From the result, DARE still outperforms HardKuma, which illustrates that the pre-trained transformer models can be employed instead. In the future, we will implement the pre-trained transformer models in other baselines and datasets.
>
> |          | |   Appearance   |      |
> | - | -| - | - |
> | Model    |     P      | R    | F    |
> | Hardkuma |     **98.8**     | 73.7   | 84.4   |
> | DARE     |     97.7     | **79.2**   | **87.5**   |
>
> **3.Discussion on disentangling rationale representation and  the non-rationale one.**
>
> Thank you for your thoughtful suggestions. First, after observing the  rationales extracted by DARE, we find most extracted rationales are continuous. In other words, the non-selected tokens are still continuous without too many non-consecutive words.
>
> Besides, we argue that your suggestion is also reasonable. Inspired by your comments, we consider that we could first disentangle the rationale representation and  the non-rationale one from the entire input text representation, as you suggested. Then, we can exploit an external guider to employ the disentangled representation to guide the rationales generation. Finally, this new method can be considered as a combination of self-guide and external guide. We will leave it as our future work.
>
> **4.The implementation information with word2vec and Bi-GRU.**
>
> For beer review prediction, we employ the  RCNN as the architecture of *selector* and *encoder* with embedding size 200 and hidden size 200.
>
> For legal judgment prediction, we adopt the Bi-GRU as the architecture of *selector* and *encoder* with embedding size 200 and hidden size 400.
>
>
>
> Reference:
>
> [1] Cheng et al. Club: A contrastive log-ratio upper bound of mutual information. ICML2020
>
> [2] Cheng et al. Fairfil: Contrastive neural debiasing method for pretrained text encoders. ICLR2021
>
> [3] Yuan et al. Improving zero-shot voice style transfer via disentangled representation learning. ICLR2021
>
> [4] Zaheer et al. Big bird: Transformers for longer sequences. NeurIPS2020

---

### Official Review · Reviewer_V57r · 2022-07-11

**Rating:** 5
**Confidence:** 4
**Soundness:** 2 fair
**Presentation:** 3 good
**Contribution:** 2 fair

**Summary:**

This paper focuses on the problem of rationale extraction for classification tasks. The proposed method addresses a hypothesized limitation in prior work, namely failing to fully exploit the original input to represent selected rationale tokens. The proposed method jointly optimizes the classification objective and a new objective which explicitly minimizes the mutual information between the representations of the selected rationales and the ones from the rest of the input to extract more information out of it.  Results on two rationale extraction tasks show improvements over existing rationale extraction methods. The developed mutual information objective is shown to be better than alternative ones on simulated experiments.


**Questions:**

The point that existing methods ignore information from the non-selected tokens in the abstract is not entirely clear at first. Since bidirectional representations are typically used to represent the tokens this statement creates confusion. It would be helpful to clarify early on that the disentanglement goal essentially aims to encourage dependence among tokens of the same type and independence for tokens of different types  (e.g. rationale/non-rationale).

Regarding the statement in lines 201-204: is the score reported with HardKuma using the paper's implementation or it is reported from prior work (line 189)? To understand the differences it is also important to know what is the number of parameters for each of them.

The evaluation is missing details on how the model and training objective hyper-parameter optimization was performed exactly. What values were examined for each hyper-parameter and what is their effect? Also, it is unclear how some design choices were made (shared encoder, MI minimization that depends on trained InfoNCE).

In Table 2, was there a reason why an experiment using plain InfoNCE with DARE was not included? It'd be useful to report performance with this objective for completeness and for clearly demonstrating the added value of the selected one.

The evaluation does not provide sufficient evidence about the quality of the extracted rationales for the second dataset since it is based on a handful of examples and t-sne. A systematic human evaluation study would be required to measure quality.

In Eq. 6, could you elaborate on how is the coherence encouraged by this regularizer? Any examples of what happens if this is not enforced would be most helpful.

**Limitations:**

Yes. There is a section in the Appendix. Limitations are not sufficiently investigated and discussed (see above).

**Strengths And Weaknesses:**

**Strengths**
- The narrative/structure of the paper was easy to follow and presentation was generally good.
- Proposes an approach for rational extraction which combines ideas from existing selection methods and objectives to minimize mutual information between selected rationales and other parts of the input.
- Presents results that show classification moderate improvements over previous rationale extraction methods on two  classification tasks.

**Weaknesses**
- Even though writing was fair in most places, some of the model design and evaluation choices lacked clarity and proper motivation/justification.
- The proposed objective to minimize mutual information appears to be incremental and the comparison between alternatives incomplete (e.g. lack of comparison with plain InfoNCE on extraction task).
- Several design choices are not well justified and the description lacks clarity in a few places (e.g. shared encoder/using trained InfoNCE in section 2.3, coherence regularizer, objective coefficients section 2.4). There are no ablation studies that demonstrate the source of improvement or contribution from each individual objective to the final performance.
- The evaluation is lacking details for understanding the results and the logic behind the design choices. Also, the quality of the extracted rationales for the judgment dataset is based only on a handful of examples and a t-sne visualization.
- The scope of the paper in the present form is limited to findings on two datasets and exploration beyond them would be required to be of interest to a broader audience.

---

> ### Author Response · Authors · 2022-07-31
> **Response to reviewer V57r**
>
> Thanks for your important comments. Below we conduct additional experiments and provide the responses:
>
> **1.Motivations of the model design.**
>
> Although existing "guidance pattern" methods achieve promising results, they fail to utilize the information of the entire text. We argue that non-rationale tokens contain rich information for rationale extraction. Specifically-
>
> Considering a text classification task as an example, in black-box approaches (e.g., GRU and BERT), we need to employ the entire text to make predictions. In contrast, the traditional rationale methods only predict based on a subsequence of the text, ignoring the semantics of the other text, which produces a certain gap with the previous black-box methods. Our DARE can exploit the ignored text well (i.e., exploiting all information of the text like the black-box approach) by encouraging the model to learn rationales while expecting it to learn what are irrelevant features.
>
> **2.Comparison with InfoNCE on extraction task.**
>
> In the disentanglement, we hope $v_{z_{1}}$ and $v_{z_{2}}$ can be independent (line 110). Therefore, we minimize their MI (min $I(v_{z_{1}} ; v_{z_{2}})$). As InfoNCE is the lower bound of MI, minimizing InfoNCE can not achieve the goal of minimizing MI. In contrast, InfoNCE is always employed to maximize MI. Therefore, we do not make a comparison with InfoNCE on extraction task.
>
> **3.Human evaluation in LJP.**
>
> We add human evaluation for the extracted rationales in LJP. Limited by time and resources, we ask four people to evaluate 100 samples for the charge prediction.
> Specifically, following [2], we evaluate rationales with three metrics: usefulness (U), completeness (C), and fluency (F), where each scored from 1 (lowest) to 5 (e.g. 2.0 and 4.9). The scoring standard for human annotators can be find in Appendix E of [2]. From the table, we find DARE outperforms HardKuma and InfoCal(HK) in all metrics.
> ||charge|||
> |-|-|-|-|
> ||U|C|F|
> |Hardkuma|3.95|3.30|3.45|
> |InfoCal(HK)|4.45|3.78|4.25|
> |DARE|**4.58**|**3.92**|**4.32**|
> **4.Experiments on the movie dataset**
>
> Thanks for your insightful suggestions. We conduct experiments on the movie dataset that is a binary classification task in Erasers [3]. As shown in the table, DARE achieves the best performance on the three metrics, where several results of baselines are quoted from [4].
> ||Movie|||
> |-|-|-|-|
> ||P|R|F|
> |Bernoulli|35.6|26.0|28.0|
> |Hardkuma|31.1|28.3|27.0|
> |InfoCal(HK)|36.7|30.3|33.2|
> |DARE|**36.9**|**32.5**|**34.6**|
>
> **5.Others**
>
> > shared encoder
>
> The goal of adopting the shared encoder is to reduce model parameters and decrease the difficulty of model training.
> > using trained InfoNCE in section 2.3
>
> Based on the Theorem3.2 in CLUB [1], we should first maximize the log-likelihood $\frac{1}{N} \sum_{i=1}^{N}\log q(y_{i}|x_{i})$ to get the trained $q(y_{i}|x_{i})$, where $q(y_{i}|x_{i})$ is the variational distribution of $p(y_{i}|x_{i})$ .
>
> In CLUB\_NCE, as the optimal critic in InfoNCE is $\log ({p(y \mid x)})$ and it is equivalent to the critic function in CLUB (i.e., the above $\log q(y_{i}|x_{i})$), we can replace the log-likelihood maximization with maximizing InfoNCE (line 141-143 in Algorithm 1).
> > coherence regularizer
>
> We take an example to illustrate how is the coherence encouraged by this regularizer, assuming there are only two tokens in the text ($x=x_{1}, x_{2}$). After applying $x$ into the selector, we get $m=m_{1}, m_{2}$ and the probability of a token not being selected $p(m_{1}=0 \mid x)$ and $p(m_{2}=0 \mid x)$.
>
> Then, there are four situations for $m$ as shown in the following table
> |m1|m2|coherence?|$p(m_{1}=0\mid x)$|$p(m_{2}=0\mid x)$|values of Eq6|
> |-|-|-|-|-|-|
> |0|0|Y|1|1|0|
> |1|1|Y|0|0|0|
> |1|0|N|0|1|1|
> |0|1|N|1|0|1|
>
> Next, we consider an extreme example where the value of $p(m_{i}=0 \mid x)$ is a binary value (i.e., $p(m_{i}=0 \mid x)=0$ or 1). After that, we can calculate Eq(6) in different situations. The results are shown in the table. Finally, we can find that in the case where $x_{1}$ and $x_{2}$ are continuous, the value of Eq(6) is 0 and vice versa. Therefore, we can conclude that minimizing Eq(6) can encourage the model to yield continuous rationales.
> >Ablation studies
>
> As the selector and predictor in DARE are based on HardKuma, we can consider HardKuma is a variant of DARE, which removes the disentanglement operation. From the result in Tbale1,2, we can find our model performs better than HardKuma.
> > HardKuma scores
>
> The score reported with HardKuma is quoted from prior work InfoCal[2] (line 189). Therefore, for fair comparison, we replicate all setups with [2].
>
> References:
>
> [1]Cheng et al. Club: A contrastive log-ratio upper bound of mutual information ICML2020
>
> [2]Sha et al. Rationalizing prediction by adversarial information calibration AAAI2021
>
> [3]DeYoung et al. ERASER: A Benchmark to Evaluate Rationalized NLP Models ACL2020
>
> [4]Yu et al. Understanding Interlocking Dynamics of Cooperative Rationalization NeurIPS2021

---

> > ### Comment · Reviewer_V57r · 2022-08-06
> > **Rebuttal acknowledged**
> >
> > Thank you for the detailed replies and clarifications. I appreciate the human evaluation of the quality of rationales; this provides more reliable evidence. It would be useful for the reader if they are included in the final version. To reflect that I increased my score.

---

> > > ### Author Response · Authors · 2022-08-07
> > > **Thank you!**
> > >
> > > We really appreciate your time and suggestions. Thanks!

---

### Official Review · Reviewer_wZNR · 2022-07-11

**Rating:** 5
**Confidence:** 4
**Soundness:** 3 good
**Presentation:** 3 good
**Contribution:** 3 good

**Summary:**

This paper aims to extraction rationales for text classification tasks. They propose to divide the sentence into two separate parts: rationale tokens and non-rationale tokens. They design a loss based on mutual information to encourage the embeddings of rationale tokens and embeddings of non-rationale tokens to be disentangled and expect more useful information can be squeezed into the rational part. Experiments and analysis are presented to support their design.

**Questions:**

Suggestions
- I feel like the reconstruction loss is needed. It takes both rationale embeddings ($v_{z_1}$ in Figure 2) and non-rationale embeddings ($v_{z_2}$ in Figure 2) as the input with the goal to reconstruct the original input. This loss encourages the model to encode as much information to $v_{z_1}$ and $v_{z_2}$ as possible while your MI loss makes $v_{z_1}$ and $v_{z_2}$ disentangled. Otherwise, without this reconstruction loss, the encoder can just learns to encode $v_{z_2}$ as random vectors. Those random $v_{z_2}$ can still have low mutual information with $v_{z_1}$, which is obviously not the expected situation. In this case $v_{z_2}$ encodes nothing and your approache will reduce the common selector + predictor approach. Although the current design somehow works and the above-mentioned case does not happen that much, I still think the reconstruction loss is needed.
- For Table 1, some of the improvements are small. Adding t-test results may help.

Questions:
- For Table 1 and Table 2, how many runs of experiments for the mean and std?
- What would happen if you also add LM regularizer just like the original InfoCal?

**Limitations:**

The authors adequately addressed the limitations.

**Strengths And Weaknesses:**

Strengths
- The writing is clear and easy to understand.
- The design is simple and seems to be effective.
- Comprehensive analysis and studies for the proposed model.

Weaknesses
- No loss to ensure that the information preserved by the non-rational embeddings are meaningful (see more details in the "Questions" section below).

---

> ### Author Response · Authors · 2022-07-31
> **Response to reviewer wZNR**
>
> We appreciate your comments! To address your concerns, below we conduct more experiments.
>
> **1.Reconstruction loss in disentanglement.**
>
> Thank you for your insightful suggestions. To this end, we have add a reconstruction loss by maximizing the MI between the original text input representations $v_{x}$ and disentanglement representations $v_{z_{1}} , v_{z_{2}}$ with  InfoNCE (i.e. maximize $I(v_{x};v_{z_{1}} , v_{z_{2}})$).
>
> We report the experimental results in the table, and so far the results are similar to DARE. One potential reason for that is the additional parameters (InfoNCE is parameterized by an MLP) which increases the difficulty of training the model. Our parameters may be not tuned to the optimum due to time constraints, we will continue our experiments in the future, thank you for your suggestion.
>
> |                          |Appearance |         |          |  Smell  |         |          | Palate  |         |          |
> | - | - | - | - | -| - | - | - | - | - |
> | Model                    | P          | R        | F | P        | R        | F        | P        | R        | F        |
> | DARE+reconstruction loss | **95.3**   | 72.9     |     82.6 | **89.8** | 46.7     | **61.4** | **85.8** | 58.8     | 69.8     |
> | DARE                     | 95.1       | **73.5** | **82.9** | 88.6     | **46.8** | 61.2     | 85.6     | **59.0** | **69.9** |
>
> **2.DARE + LM regularizer.**
>
> As our DARE is focused on improving the performance of extracted rationales though the disentanglement, no regularizer changes have been made. As per your suggestion, we add the LM regularizer proposed by InfoCal [1].
>
> From the result, we observe that DARE + LM regularizer performs better than the original InfoCal, which demonstrates the effectiveness of our disentanglement operation.
>
> |                     |Appearance |         |          |  Smell  |         |          | Palate  |         |          |
> | ------------------- | ---------- | -------- | ------- | -------- | -------- | -------- | -------- | -------- | -------- |
> | Model               | P          | R        |        F | P        | R        | F        | P        | R        | F        |
> | InfoCal             | **98.5**   | 73.2     |     84.0 | 95.6     | 45.6     | 61.7     | **89.6** | 59.8     | 71.1     |
> | DARE                | 95.1       | 73.5     |     82.9 | 88.6     | 46.8     | 61.2     | 85.6     | 59.0     | 69.9     |
> | DARE+LM regularizer | 96.2       | **75.5** | **84.6** | **90.2** | **50.1** | **64.4** | 87.6     | **60.5** | **71.6** |
>
> **3.How many runs of experiments for the mean and std?**
>
> We run five time experiments for the mean and std in Table 1 and Table 2.
>
> Reference:
>
> [1] Lei Sha, Oana-Maria Camburu, and Thomas Lukasiewicz. Learning from the best: Rationalizing prediction by adversarial information calibration. AAAI2021

---

> > ### Comment · Reviewer_wZNR · 2022-08-07
> > **Thanks for the response**
> >
> > Thanks for your response. For (1), I guess it can be hard to train when adding reconstruction loss. Another possible reason can be that your original loss somehow encourages model to capture key information (rationales) well, and disentanglement may be not the rooted factor and not necessary for the improvement. This just a guess, maybe conducting hyper-parameter search can prove that reconstruction loss works.
> >
> > I have no other questions, thanks!

---

> > > ### Author Response · Authors · 2022-08-08
> > > **Thank you!**
> > >
> > > Thank you for your comments!
> > >
> > > We argue that a possible reason why the reconstruction loss does not work is that it is hard to train, where there are many losses involved (line 158).
> > >
> > > Below we discuss your proposed another possible reason.
> > >
> > > **Comments**: Another possible reason can be that your original loss somehow encourages model to capture key information (rationales) well, and disentanglement may be not the rooted factor and not necessary for the improvement
> > >
> > > **Responses**: We argue that disentanglement is necessary for the improvement. To validate it, we make an ablation study which removes the disentanglement operation in DARE.
> > >
> > > Specifically, as the selector and predictor in DARE are based on HardKuma,  we can consider HardKuma is a variant of DARE, which ablates the disentanglement operation
> > >
> > > From the result in Table 1,2, we can find our model performs better than HardKuma. Therefore, we believe that disentanglement is necessary for the improvement.
> > >
> > > In the future, we will conduct the hyper-parameter search to validate the effectiveness of the reconstruction loss, and we will add this experiment in our revision paper.

---

> > > > ### Comment · Reviewer_wZNR · 2022-08-09
> > > > **Thanks**
> > > >
> > > > Thanks for your reply! I am looking forward to the results of reconstruction loss.

---

### Official Review · Reviewer_1CAu · 2022-07-18

**Rating:** 6
**Confidence:** 4
**Soundness:** 3 good
**Presentation:** 2 fair
**Contribution:** 3 good

**Summary:**

This work aims at proposing a rationale extraction method that disentangles the rationale and non-rationale spans. Specifically, apart from predicting the label from the rationale span, the proposed method DARE adds another main objective that minimizes the mutual information (MI) between the representations of the rationale and non-rationale spans. Modified upon previous MI minimization work InfoNCE and CLUB, the authors propose a new method CLUB_NCE and use it in DARE. Experiments on two natural language datasets, BeerAdvocate and Legal Judgment Prediction show a performance advantage of DARE over the baselines.

**Questions:**

Please see my concerns in the previous section. Also, RCNN is used as the selector and encoder architecture. Can pretrained transformer models be used instead? If so, would the re-embed operation in Sec 2.2.2 (line 101) lead to any issues?

**Strengths And Weaknesses:**

The paper presents an overall novel and effective method for rationale extraction. The comparison between different MI minimization methods is aided by simulated experiments. The proposed method DARE is introduced rather clearly. Experiments on two natural language datasets show advantages of DARE over a few baseline methods.

The concerns that some readers might raise and complain include:
1- At line 88 and 111, a main argument for adding a regularizer that minimizes the MI between rationale and non-rationale representations, is that it can make the rationale more “comprehensive” and “rich” information. Can this argument be supported more intuitively with a discussion? (the connection between A and B’s disentanglement and A being “comprehensive”)

2- In Sec 2.3.2, CLUB_NCE is mainly CLUB with the approximation network p(y|x) replaced by the InfoNCE scoring function. While there are performance gains as shown in the experiments, the potential motivation and rationale behind this modification is not fully clear and can benefit from a further discussion.

3- The experiment presentation in Sec 3.2.2 needs improvements. The description and figure for the t-SNE experiment look confusing. How were the blue and red circles selected in a formal way? Can the distance between clusters be measured? The scale of 5(a) and 5(b) is different, making it harder for a comparison.

4- At line 295, the L1Out method was in previous tables but was not introduced until this stage. This should probably appear earlier in the text with more descriptions.

---

> ### Author Response · Authors · 2022-07-31
> **Response to reviewer 1CAu**
>
> Thank you for your time and insightful suggestions! According to your comments, we conduct additional experiments and provide the responses as follows:
>
> **1.Can the disentanglement argument operation make the rationale more comprehensive and rich information?**
>
> Although existing "guidance pattern" methods achieve promising results, these methods fail to utilize the information of the original text. We argue that disentangled non-rationale tokens contain rich information for making rationales more comprehensive. Specifically--
>
> (1) Considering a text classification task as an example, in black-box approaches (e.g., GRU and BERT), we need to consider the semantics of the entire text input to make predictions. In contrast, the traditional rationale approach only predicts based on a subsequence of the entire text input, ignoring the semantics of the other text inputs, which produces a certain gap with the previous black-box approaches. Our DARE can exploit this ignored text input well (i.e., like the previous black-box approach, making full use of all the information in the text) by encouraging the model to learn rationales while expecting it to learn what are irrelevant features.
>
> (2) When generating the rationales, adjacent tokens are encouraged to be selected for achieving fluency of rationales by the regularizer $\mathcal{L}_{re}$. However, this may lead to the selection of unnecessary tokens due to their adjacency to informative ones [1]. For example, in Figure 1, the rationale extracted may be "*. The defendant slashed the victim with a knife, which caused a serious injury to the victim.*", where "*. The*" may be selected due to their adjacency. As our DARE encourages the model to disentangle the rationale tokens and non-rationale tokens, this problem can be alleviated.
>
> **2.The motivation and rationale of CLUB\_NCE.**
>
> $p(y|x)$  in CLUB can be any neural network. Intuitively, an easy way to parameterize $p(y|x)$ is with an MLP. However, in practice,  $p(y|x)$ is always parameterized by a Gaussian family [2,3,4], where one potential reason for not employing MLP is that MLP is difficult to converge in CLUB. Therefore, it is significant to design a function that can parameterize $p(y|x)$  with any network and be easy to converge.  As the critic function in InfoNCE can be any network, and the optimal critic $\log ({p(y \mid x)})$ is equivalent to the critic function in CLUB. Empirically, we combine CLUB with InfoNCE.
>
> **3.Further explanation for Figure 5.**
>
> Specifically, For Figure 5(a), the blue circle selected represents the tokens that are predicted correctly by HardKuma; and the red circle selected denotes the tokens that are predicted incorrectly.
>
> For Figure 5(b), the blue circle selected represents the tokens that are predicted correctly by DARE; and the red circle selected denotes the tokens that are predicted incorrectly by HardKuma, where our DARE identify that these tokens are non-rationale tokens. From Figure 5(b), we can find the tokens predicted incorrectly by HardKuma are accurately predicted to be non-rationale tokens.
>
> Besides, we report the Euclidean distance between the blue circle and the red circle in Figure 5(a) and 5(b).
>
> |      | Figure5 (a) | Figure5 (b) |
> | ---- | ----------- | ----------- |
> | EU   | 7.62        | 12.18       |
>
> From the table, we can find the EU distance in Figure5 (b) is higher than it in Figure5 (a), which demonstrates that DARE can well disentangle rationale representations.
>
> **4.Discussion on pretrained transformer models.**
>
> First, we argue that the pretrained transformer models can be used instead. We report the results in Appearance with HardKuma and DARE, which both implement BigBird [5] to replace RCNN.
>
> From the result, DARE still outperforms HardKuma, which illustrates that the pretrained transformer models can be employed instead. In the future, we will implement the pretrained transformer models in other baselines and datasets.
>
> Besides, the re-embed operation in the pretrained transformer models can be solved by setting attention_mask (see huggingface[ https://huggingface.co/docs/transformers/index ] for detail) as $m$, indicating which tokens can input the self-attention layer in the transformer.
>
> |||Appearance||
> |-|-|-|-|
> |Model|P|R|F|
> |Hardkuma|**98.8**|73.7|84.4|
> |DARE|97.7|**79.2**|**87.5**|
>
> **5.The position of descriptions of L1Out method.**
>
> Thank you for your suggestions. We will describe L1Out earlier in the text in our revision paper.
>
> **Reference:**
>
> [1] Sha et al. Learning from the best: Rationalizing prediction by adversarial information calibration. AAAI2021
>
> [2] Cheng et al. Club: A contrastive log-ratio upper bound of mutual information. ICML2020
>
> [3] Cheng et al. Fairfil: Contrastive neural debiasing method for pretrained text encoders. ICLR2021
>
> [4] Yuan et al. Improving zero-shot voice style transfer via disentangled representation learning. ICLR2021
>
> [5] Zaheer et al. Big bird: Transformers for longer sequences. NeurIPS2020

---

### Author Response · Authors · 2022-08-01
**General response**

We sincerely appreciate all reviewers' time and efforts in reviewing our paper. We would like to thank all reviewers for providing many insightful and valuable suggestions. Below is our summary of the responses:

1.**More discussions**: We justify the motivation and rationale of the disentanglement and CLUB\_NCE (1CAu, GAz6). Also, we provide intuitional examples for key concepts (V57r). Besides, we further describe experimental results (1CAu). Finally, we discuss improvements in model design (wZNR).

2.**More Experiments**: We make experiments to validate the feasibility of employing the pre-trained model instead of the current encoder (1CAu, GAz6). Also, we make a human evaluation and add a new dataset to further demonstrate the effectiveness of DARE (V57r).

We will add detailed discussions and experiments to our paper in the future.

We hope our responses can clarify all your confusion and alleviate all concerns. We thank all reviewers again. Looking forward to your reply!

---

### Public Comment · ~Francois_Barnard1 · 2023-06-22
**CAIL2018 in English**

I saw in your paper that your results are in English. I searched online, and it appears that CAIL2018 is Chinese only (https://huggingface.co/datasets/cail2018). It would be great if the authors could point me towards an English source or describe how they arrived at the English version of CAIL2018.

Thanks you

---

> ### Public Comment · ~Linan_Yue1 · 2023-06-23
> **Response to Francois Barnard**
>
> Hi, the full name of CAIL data is Chinese AI and Law challenge and it only focuses on Chinese law. The English results we show in the paper about CAIL are translated into English based on the corresponding Chinese results.
>
> Best wishes.
>
> Linan Yue

---

### Meta-Review · Area_Chair_rHVd · 2022-08-25

**Recommendation:** Accept
**Confidence:** Certain

**Metareview:**

This paper presents an improved method for improving rationale extraction and model explainability. The key insight is to disentangle rationale from non-rationale tokens in the input. Reviewers raised some fairly minor questions regarding the notion of disentanglement, the proposed loss, and experiments (e.g, comparison with related models). The authors responded to the queries and included more discussion and results. The paper is solid, the results look good, and the added discussion will clarify the raised concerns.

**Award:**

No

---

### Decision · Program_Chairs · 2022-09-14

Accept